# Processed Food Additive Microbial Transglutaminase and Its Cross-Linked Gliadin Complexes Are Potential Public Health Concerns in Celiac Disease

**DOI:** 10.3390/ijms21031127

**Published:** 2020-02-08

**Authors:** Aaron Lerner, Torsten Matthias

**Affiliations:** AESKU.KIPP Institute, Mikroforum Ring 2, 55234 Wendelsheim, Germany; matthias@aesku.com

**Keywords:** celiac disease, microbial transglutaminase, food additive, cross-linking, immunogenicity, pathogenicity

## Abstract

Microbial transglutaminase (mTG) is a survival factor for microbes, but yeasts, fungi, and plants also produce transglutaminase. mTG is a cross-linker that is heavily consumed as a protein glue in multiple processed food industries. According to the manufacturers’ claims, microbial transglutaminase and its cross-linked products are safe, i.e., nonallergenic, nonimmunogenic, and nonpathogenic. The regulatory authorities declare it as “generally recognized as safe” for public users. However, scientific observations are accumulating concerning its undesirable effects on human health. Functionally, mTG imitates its family member, tissue transglutaminase, which is the autoantigen of celiac disease. Both these transglutaminases mediate cross-linked complexes, which are immunogenic in celiac patients. The enzyme enhances intestinal permeability, suppresses mechanical (mucus) and immunological (anti phagocytic) enteric protective barriers, stimulates luminal bacterial growth, and augments the uptake of gliadin peptide. mTG and gliadin molecules are cotranscytosed through the enterocytes and deposited subepithelially. Moreover, mucosal dendritic cell surface transglutaminase induces gliadin endocytosis, and the enzyme-treated wheat products are immunoreactive in CD patients. The present review summarizes and updates the potentially detrimental effects of mTG, aiming to stimulate scientific and regulatory debates on its safety, to protect the public from the enzyme’s unwanted effects.

## 1. Introduction

The use of food additives is soaring in the processed food industry. Some are labeled, although many which have been characterized by the FDA as GRAS (generally recognized as safe) are not. GRAS implies that scientifically qualified experts agreed that data generated by the manufacturer are adequate and safe concerning a compound’s intended use. In the last decade, scientific and public concerns were raised as to the adequacy, efficiency, and reliability of the current policy of the regulatory authorities as to food safety [1,2]. Even the US Government Accountability Office (GAO), reporting to congressional requesters, criticized the GRAS process [3]. Furthermore, major concerns have been raised, most recently, as to the GRAS determination process in children. The council on environmental health announced in 2018 that “Substantial improvements to the food additives regulatory system are urgently needed, including greatly strengthening or replacing the “generally recognized as safe” (GRAS) determination process, updating the scientific foundation of the FDA’s safety assessment program”. They even urged “retesting all previously approved chemicals and labeling direct additives with limited or no toxicity data” [4,5].

Based on the above background and updated scientific knowledge, the present review will focus on an unlabeled, GRAS defined, heavily used food additive, i.e., microbial transglutaminase (mTG). 

This review is not intended to systemically cover the entities of mTG or its nutritional applications. Instead, this targeted article will present a narrative review focusing on a niche topic, namely the cross-talks between mTG, its gliadin cross-linked complexes, and celiac disease (CD). 

### 1.1. Microbial Transglutaminase: Features and Functions in Food Industries

Transglutaminase (EC 2.3.2.13), i.e., protein-glutamine γ-glutamyltransferase, belongs to the family of transferases. It catalyzes the formation of a covalent isopeptide bond between peptidyl bound glutaminyl resides (acyl donor), and first-order ε-amine groups of multiple compounds, for instance, proteins/peptides (acceptors of an acyl residue). mTG is a heavily used enzyme in the processed food industry [6,7]; it is intended to improve products’ properties by catalyzing the cross-linking of proteins. It is considered as a food/meat glue and used routinely in meat, fish, seafood, surimi, milk and dairy, sweet foods, and bakery products [8,9,10,11]. mTG improves dough stability, elasticity, persistency, volume, hydration, elongates shelf life, and provides better fermentation tolerance in baked goods such as bread, pastries, pasta, and tortillas. The enzyme is also used in gluten-free products to improve their texture and qualities, as announced by manufacturers. As a result, Western societies ingest mTG in substantial quantities [6,12].

### 1.2. Enteric Luminal Sources of Microbial Transglutaminase

As a universal and pleiotropic enzyme that is ubiquitously and abundantly expressed, mTG is found in many prokaryotes and eukaryotes. The extra- and intra- luminal sources of mTG were summarized recently [7,12,13,14]. Table 1 updates those sources, adding “viral transglutaminase” [15,16,17,18]. Interestingly, when studying the amino acid sequences of the recently described *prevotella*-specific megaphages, a transglutaminase-like sequence was found in human and several animal stool samples when aligned with mTG amino acid sequence (Ramesh Ajay unpublished personal communications) [19]. Examining the phylogenetic tree of the mTG compared to the phage transglutaminase-like sequences, it seems that the mTG and the bacteriophage transglutaminase-like sequences evolved from an ancestral creature. Along the evolution, they split apart, resulting in more homology between humans and pigs, as compared to the mTG (Ramesh Ajay unpublished personal communications). 

Since mTG is increasingly used as a food additive during food processing [6,12,20], several facts should be mentioned. Each kilogram of food treated with mTG contains about 50–100 mg of mTG. The average yield of *Streptoverticillium mobaraense* used by several food industries for mTG production is 1.25 unit/mL or 22 unit/mg. The average Western diet contains large amounts of mTG, with an estimated maximum daily intake of up to 15 mg [8,12,21,22,23]. In summary, it appears that a substantial amount of mTG activity occurs in the human enteric lumen.

### 1.3. Celiac Disease in a Nutshell

Celiac disease is part of the autoimmune inflammatory disease family targeting the small intestine. The disease is triggered by the ingestion of the prolamins contained in wheat, barley, rye, or oat, by genetically susceptible individuals [39]. The average incidence of CD is 1% in the Western world. The majority of patients are a/hyposymptomatic, being undiagnosed. The autoantigen in CD is the enzyme tissue transglutaminase (tTG), that functionally imitates its family member mTG [12,13,14]. Both transaminases can deamidate or cross-link (transamidate) gliadin peptides, thus potentiating their attachment and presentation by the HLA-DQ2/8 grooves to stimulate committed celiacogenic CD4+ T cells, inducing mucosal inflammation and destruction [40,41]. The epidemiology and phenotype of CD are constantly changing. In recent decades, there was an epidemiological shift in the presenting phenotype toward a more advanced age, and an increased prevalence of latent, a/hyposymptomatic behavior [42]. The only acceptable and proven therapy is lifelong adherence to a gluten-free diet (GFD). However, during adolescence and subsequently, compliance is decreasing, reaching 60% noncompliance during adulthood and old age [43]. Despite the beneficial effects of the GFD, it poses many difficulties accompanied by social pressure [44,45]. On the other hand, many patients remain symptomatic and have ongoing low-grade enteric inflammation, despite gluten withdrawal. Admittedly, in contrast, gluten has its known side effects [46], but GFD also has a dark side [47]. 

The question of whether additional environmental factors affect CD progression remains unraveled. Various environmental factors have been suggested as inducers. [48]. Most recently, the leaky gut theory was suggested, whereby multiple industrial food additives increase intestinal permeability, resulting in autoimmune disease induction [6]. Given the uncertainty regarding causality, these associations between CD and environment are yet unclear. Further investigations are required to elucidate the potential mechanisms by which modern exposures contribute to CD induction and progression. The present review summarizes the theoretical background and the available scientific data, and puts forward the hypothesis that mTG might be the missing link [12]. It might explain the surge in CD incidence, witnessed in recent decades [49,50], the changing epidemiology and the presenting symptoms, and the selected patients that do not improve on a GFD. 

## 2. Microbial Transglutaminase-Gliadin Cross-Linked Complexes Are Immunogenic in Celiac Disease

Gliadin peptides are ideal substrates for transglutaminases, be it the tTG or the mTG [7,12,13,14,24,25,40,41]. Natively, the TGs catalyzes the transamidation of peptidyl bound glutaminyl resides (acting as an acyl donor) with a primary amine, typically the side chain of lysine residue (acting as an acyl acceptor), resulting in protein cross-linking. tTG/mTG gliadin cross-linked complexes are created, and neo-epitopes appear on the complexes’ surfaces. The transition from naive antigens to foreign ones represents a loss of tolerance, resulting in reactive immune stimulation and the generation of a new family of antibodies, namely, neo-epitope antibodies. Neo-epitope tTG (tTG neo) [14,25,51,52,53,54,55] and neo-epitope mTG (mTG-neo) antibodies [7,13,14,25,51,52,53,54,56,57] appear in the systemic circulation. The present review will concentrate on the immunogenicity of the mTG-gliadin cross-linked complexes, mTG-neo antibodies in CD. 

The mTG-gliadin cross-linked complexes are immunogenic in CD. Several recent studies have compared mTG-neo antibodies with other CD associated antibodies. Anti-mTG, tTG, mTG-neo, and tTG neo were studied in 95 pediatric celiac patients, 99 healthy children, 79 normal adults, and 45 children with nonspecific abdominal pain [58]. mTG-neo IgG sensitivity, specificity, negative predictive value, positive predictive value, and Area under the Curve (AUC) were 94.9%, 93.9%, 94.9%, 94.0%, and 0.94%, respectively. It was concluded that mTG is immunogenic in children with CD. Upon complexing with gliadin, its immunogenicity is further enhanced. Using Marsh criteria, anti-mTG-neo IgG antibodies correlated with intestinal damage to a comparable degree as the anti-tTG-neo IgA. mTG and tTG displayed a comparable immunopotent epitope. mTG-neo IgG was declared as a new marker for CD diagnosis [58]. 

A more recent back to back comparison evaluated 17 different serological markers of CD, including mTG-neo isotypes [51]. This time, 95 CD patients were compared to pathological and healthy controls.

mTG neo IgG had 95.9%, 93.9%, 95.8%, 94.0%, 0.95%, sensitivity, specificity, negative predictive value, positive predictive value, and AUC, respectively. mTG-neo IgG titers followed the tTG-neo IgA+G performance. Once again, mTG-neo IgG was found as a new serological biomarker for CD [51]. In another study, 296 Swedish children with untreated CD and 215 nonceliac disease controls were checked for 10 CD associated antibodies [52]. Although the tTG-neo IgA+G had the best performance, the mTG-neo IgA+G had good ones. The sensitivity, specificity, and Area under the Curve were 75.3%, 88.8%, and 0.903, respectively. Comparing the mTG-neo isotypes, the combined mTG-neo IgA+G was the most useful to reflect the enteric damage, as defined by Marsh criteria. High levels of anti-mTG-neo IgG and anti-tTG-neo IgG were present in the earlier age groups, as compared to anti-tTG IgG (*p* < 0.001). Considering the antibody diagnostic performance (based on AUC, enteric damage reflection, and predictability at an early age), the combined mTG-neo check had good results. This might represent a new marker for CD screening, diagnosis, and predictability, although not the best one [52]. Better designed and more controlled studies are requested. Recently, mechanisms have been suggested for the complexes’ immune pathways. When mTG and gliadin are gold tagged and examined by an electron microscope, it can be observed that they internalize into enterocytes. They are transmitted through the early–late endosomes into the endoplasmic reticulum and deposited in the basolateral areas [59]. The authors conclude, “The strong localization of mTG at the basolateral membrane and the lamina propria may also indicate a potential antigenic interaction with cells of the immune system”. In other words, codeposited mTG and gliadin, nonself-antigens, are facing the most crowded immune systems. Most probably, activating the reactive branch, resulting in mTG-neo antibodies secretion. It should be emphasized that the covalent isopeptides created during the cross-linking of the gliadin peptide are very stable and resistant to endogenous/exogenous proteases, detergents like bile salts, temperature, and a large range of pH [12,13,14,56]. It can be summarized that the mTG-gliadin cross-linked complexes are resistant, and that their components, transmitted through enterocytes, are immunogenic in CD. 

## 3. Microbial Transglutaminase Is Potentially Pathogenic to CD Patients

The potential pathogenic mechanisms connecting the mTG to CD pathogenesis were summarized recently [7,12,13,14,25,27,56,58]. The following paragraphs will update and add some additional pathways for the mTG-CD cross-talks.

### 3.1. Enhancing Intestinal Permeability. 

Several pathogenic mechanisms can be envisioned: 

a. mTG is a survival mechanism for multiple intestinal lumen inhabitants, members of the microbiome, dysbiome, or the pathobiome (Table 1). On the other hand, infections are a powerful disruptor of tight junction functional integrity [60]. 

b. Gluten and gliadin peptides are known to increase intestinal permeability, not only in gluten-dependent diseases, but also in CD familial first-degree relatives, and even in the normal population [61,62]. Since mTG cross-link gliadin peptides, the gluten part of the complexes may, by itself, open gut permeability. 

c. Essential functional tight junction proteins contain peptidyl bound glutaminyl resides (acyl donors) and prolines (acyl acceptors) in their sequences. Claudins, occludins, F-actin, myosin, zonulin, F-cadherin, catenin, keratin are some examples. They represent ideal substrates for mTG mediated cross-linking, and thus, may compromise gut permeability. 

d. mTG has emulsifying properties [25]. The use of emulsifiers is increasing in the processed food industry, and they are known to enhance intestinal permeability [6,63,64,65,66]. Nanoparticles and mTG have multiple industrial applications in the processed food domain [6,67,68]. By opening tight junctions, they affect health safety [6,69]. 

e. Glutamine and zinc deprivation alter intestinal permeability [70,71,72]. Since mTG cross-links glutamine residues, it might result in glutamine deficiency. Its deprivation can increase enteric permeability [71,72]. 

f. Histones are crucial in suppressing or inducing gene expression. They are cross-linked by transglutaminase, and epigenetics is important in CD pathophysiology [73,74,75,76]. Being a substrate of mTG, histone cross-linking might reduce its availability. 

Since intestinal permeability is compromised in CD, the question arises as to which part mTG or mTG-gliadin cross-linked complexes are playing a role in the leaky gut in CD?

### 3.2. Suppression of Mechanical and Immunological Enteric Protective Barriers

Efficient barriers evolved during human evolution to protect us against noxious environmental factors. The mTG gut cargo can affect such mechanical and immune barriers. The mucus cohabits the upper part of the luminal epithelium all along the gastrointestinal tract. It protects the enteric epithelium from microbial penetration and adherence to the epithelial surface. Its main structural compound is MUC2 mucin, which is rich in glutamine and lysine residues. tTG has affinity to the MUC2 CysD2 domain, catalyzes its transamidative cross-linking, and thus enhances its antibacterial protective function [77]. Luminal mTG may compete with the endogenous tTG, thereby altering mucus stability, and enabling the intestinal dwellers to attach to their corresponding receptors. It is the correct place to emphasize that mTG/tTG isopeptide-bonds are incredibly resistant to luminal proteases, reducing agents, detergents, bile acids, immunoglobulins, and other antimicrobial molecules [13,25,78,79]. Secondly, *Streptococcus suis* mTG was described recently to exert antiphagocytic activity, thereby compromising a powerful and indispensable immune protective mechanism [26,80,81]. The same *Streptococcus suis* can evade neutrophils by preventing neutrophil extracellular trap formation. However, the mTG involvement in this phenomenon is not yet apparent [82,83]. 

### 3.3. Enhancing Bacterial Growth

mTG is capable of improving the growth performances of microbes. When the *Streptoverticillium mobaraense* mTG gene was cloned into *Lactococcus lactis*, a significantly higher microbial biomass was shown [84,85]. The authors suggested that the industrial food-grade mTG expression increased intracellular pH, thus saving ATP energy, that was redirected for growth, thereby enhancing microbial proliferation [84,85].

### 3.4. Increasing Uptake of Gliadin Peptide

tTG facilitates the apical-basal transfer of gliadins helped by apical transferrin receptors and secretory IgA [86]. Furthermore, gliadin uptake is facilitated by transglutaminase applications on the cell line in vitro [12,25]. By imitating those functions, mTG can potentially facilitate this epithelial gliadin uptake pathway, thereby enhancing CD. 

### 3.5. mTG and Gliadin Molecules are Cotranscytosed through the Enterocytes

Recently, Stricker S. et al. documented mTG and gliadin peptide transport into CD enterocytes and a RACE cell line (a special cell type called Rapid uptake of Antigen into the Cytosol of Enterocytes) [59]. The two environmental antigens were transported by the endoplasmic reticulum, to be deposited at the basolateral membrane of the enterocyte. The pivotal study is a proof of concept that luminal mTG is capable of translocating below the epithelial layer. The enzyme comigration with a gliadin peptide further reinforces the ideal place and circumstances for mTG-gliadin cross-linked subepithelial deposition. In fact, subepithelial deposits involving IgA-tTG are among the hallmark markers for early CD [87,88]. Unfortunately, no study has yet explored mTG cross-linked gliadin in those deposits.

### 3.6. Potential mTG–Gliadin Uptake by Mucosal Dendritic Cells

Various sensing cells and surface molecules survey the luminal content. One of these is the lamina propria dendritic cells. Situated among the enterocytes or elongated between them, they are ideally situated to sense the luminal compartment [89,90,91]. Being efficient in antigen-acquisition, processing, and presenting cells, they play crucial roles in inducing protective immunity or tolerance towards luminal dwellers or constituents. Interestingly, the duodenal mucosa of CD patients expresses tTG on their cell surface. Raki and others have shown evidence that monocyte-derived dendritic cells endocytose this surface-associated tTG [92,93]. A substantial amount of mTG and partially digested gliadin peptides reside in the proximity of the enterocyte’s brush border. This might represent a new port of entry for mTG and gliadin monomers or cross-linked complexes to face the local subepithelial immune systems. The intraepithelial dendritic cell’s surface transglutaminase transcytosis complements the enterocytic endocytosis of mTG and gliadin peptides recently described by Stricker et al. [59]. The place of the other sensing mechanisms of the luminal content, namely, M cells, colonocytes, enteroendocrine, and enteric glial cells, are far from being elucidated in mTG/gliadin peptide uptake in CD. 

### 3.7. mTG Treated Wheat Products Are Immunoreactive 

The post-translational modification of wheat proteins exerted by mTG-induced transamidation [24,25] has been checked in humans. It appears that mTG treated wheat/gluten/gliadin products induces various immune reactions: 

1. Reaction with IgA-anti gliadin antibodies [94].

2. Recognition by IgA from CD patients in an age-dependent manner [95].

3. Induction of intestinal interferon-c release and serum tTG and endomysial antibodies in CD patients [96].

4. IgA reactivity of CD sera was higher against gluten-containing compared to gluten-free mTG treated bread. Surprisingly, the electrophoretic pattern of gluten-free bread prolamins was changed by the mTG treatment, and 31,000 new bands originating in maize were recognized by some CD patients’ IgA. [97].

5. mTG treated wheat products are immunoreactive [98,99]. 

6. The mTG treated products recognition by gluten-specific T cells, thus, increasing gluten immunoreactivity [100].

The debate of using mTG transamidating capacity to produce safer gluten-free products as a future therapeutic strategy for CD is an open one; at present, it is lacking successful CD patient double-blind, cross-over challenges [101,102,103,104,105,106].

## 4. Should Microbial Transglutaminase Be Declared as a Public Health Caveat in Celiac Disease?

As declared by the manufacturers, mTG and its cross-linked products are considered safe, nontoxic, nonimmunogenic, nonallergenic, and nonpathogenic for public health [12]. The present narrative review summarizes the epidemiological, scientific, serological, and clinical proof for the use of mTG and its transamidated products as food additives, as well as theirimmunogenicity, and potential pathogenicity. There is enough background knowledge to address mTGs’ safety in a multi-disciplinary approach. Protection of public health against the adverse effects of mTG should be of the utmost importance. The FDA and the European Food Safety Authority recently published the regulatory demands and follow-ups on the GRAS status of food additives and microbial enzyme usage by the food industries [107,108,109,110]. Because of the enzyme mTG and its gliadin cross-linked complexes’ potential pathogenic activities, summarized in Figure 1, the authors urge the worldwide food safety regulatory authorities to analyze the current knowledge and consider the alleviation of the GRAS status from this heavily used food additive. Better early than late, as Benjamin Franklin sited: “An ounce of prevention is worth a pound of cure.” If withdrawn, these findings will change food labeling, food additive policies, and regulatory product control, thus affecting consumer health and safety.

## 5. Conclusions

The processed food additive mTG and its cross-linked gliadin complexes are potential public health concerns in CD. Functionally, the enzyme imitates **its family member tTG**, the autoantigen of CD. Both enzymes mediate cross-linked complexes, which are immunogenic in celiac patients. mTG enhances intestinal permeability, suppresses mechanical and immunological protective barriers, increases luminal bacterial growth, and augments the uptake of gliadin peptide. mTG and gliadin molecules are co-transcytosed through the enterocytes and are deposited sub-epithelially. More so, the enzyme-treated wheat products are immunoreactive in CD patients. It is hoped that the present review will encourage clinical, scientific and regulatory debates on mTG safety, to protect the public from the enzyme’s unwanted effects.

## Figures and Tables

**Figure 1 ijms-21-01127-f001:**
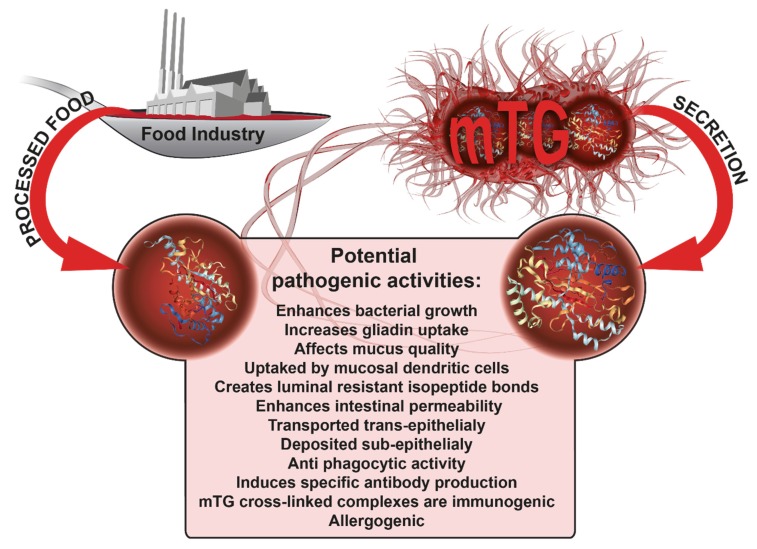
The potential immunogenic and pathogenic activities of mTG and its gliadin cross-linked complexes in CD evolvement. mTg-microbial transglutaminase, CD-celiac disease.

**Table 1 ijms-21-01127-t001:** The extra- and intra- intestinal sources of mTG or tTG-like activity in the human gut. (Adapted from ref. [7,12,13,14]). mTG-microbial transglutaminase, tTG-tissue transglutaminase.

MTG Source		Reference
**Extra-Intestinal**	Processed food additive	[6,8,9,11,12,13,14,20]
	Pathobionts	[24,25,26]
	Probiotics	[27,28]
	Plants	[29,30]
	Vegetables	[31]
	Meat	[32,33]
**Intra-Intestinal**	Microbiome	[7,12,13,14,24,25]
	Dysbiome	[7,12,13,14,24,25]
	Yeasts	[34,35,36]
	Fungi	[37,38]
	Viruses	[15,16,17,18]

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
