# Peer review of "Processed Food Additive Microbial Transglutaminase and Its Cross-Linked Gliadin Complexes Are Potential Public Health Concerns in Celiac Disease"

_ijms, 2020, doi:10.3390/ijms21031127_

Round 1

Reviewer 1 Report

This is a review focused on microbial transglutaminase (mTG), a food additive with some similarities with tissue transglutaminase (tTG), but which is “generally recognized as safe”. In this review, Authors questioned the safety of mTG and discussed about the interaction with gliadin and subsequent potential implications in the immunogenesis of celiac disease (CD).

It is a very interesting review that unveils a little known aspect of CD immunology. I have only a few minor objections:

1) Referring to the histological damage in case of positive mTG neo-IgG (lines 159-162), how was graded histology damage? Did villous atrophy appear or only IELs increase? Please give more details about such studies.

2) Can the onset of mTG neo-IgG explain some cases of CD refractory to gluten free diet?

3) Can Authors roughly estimate the safety threshold of mTG daily intake?

Author Response

To reviewer No 1:

Thanks for the complements. The text was extensively change and the English improved .all changes are marked in Yellow

1) Referring to the histological damage in case of positive mTG neo-IgG (lines 159-162), how was graded histology damage? Did villous atrophy appear or only IELs increase? Please give more details about such studies. The damage was graded according to Marsh criteria. Please, see lines; 160, 175. CD was diagnosed only with positive serology and Marsh ≥2. Marsh 1 was not diagnosed as celiac disease.

2) Can the onset of mTG neo-IgG explain some cases of CD refractory to gluten free diet? A very interesting question, but, was never checked, but, theoretically yes. Should be studied in the future

3) Can Authors roughly estimate the safety threshold of mTG daily intake? We can’t since it was not evaluated. There are several other sources of mTG in the gut lumen (table 1).

Reviewer 2 Report

I thought it started well but fragmented as it went along. There are quite a few colloquialisms, a tautology and some issues with in text referencing style. 

In page 2 the use of "were" should be either "are" or "were/are"

Be consistent with Western - you use western and Western

Line 20 - insert "therefore"

Remove No when referring to tables. Capitalise T in tables.

The section 2 lines 163-178 need a rewrite - it is difficult to follow.

Section 3.1 has a fragmented feel and the section c is hard to follow. 

section e "it is resulting" scans badly

line 218 "and epigenetic is vital" does not scan.

You sometimes refer to glutamine side chains instead of glutaminyl side chains (a major context error)

There is a semi-repeat in line 232-233

In 3.3 the pH change effects are not well described

In line 250 "On the same line" sounds clumsy and colloquial.

In 3.5 RACE is not explained 

267 - contains your tautology of "extremely crucial".

289 - Finally. - at the end of the list seems clumsy. Also most of the list seems poorly written. For example Recognised should be "Recognition of"

There are several examples of terms that would benefit from addition of "s" - like induce (induces) in line 282. 

Overall - I think you have an interesting subject and it seems to update some earlier work - but it needs a general tidy up and greater integration (avoid lists). 

With more editing I think it could be worthy of publication but currently is at too early a stage for this.

Author Response

To reviewer 2

Thanks for the valuable review. The manuscript was extensively changed and upgraded. All changes are marked in Yellow.

There are quite a few colloquialisms, a tautology and some issues with in text referencing style. Please, see the changes in yellow, along the text.

In page 2 the use of "were" should be either "are" or "were/are" Corrected, lines 41, 42.

Be consistent with Western - you use western and Western Corrected

Line 20 - insert "therefore" inserted

Remove No when referring to tables. Capitalise T in tables. Done

The section 2 lines 163-178 need a rewrite - it is difficult to follow. Done

Section 3.1 has a fragmented feel and the section c is hard to follow. Improved

section e "it is resulting" scans badly.  Corrected

line 218 "and epigenetic is vital" does not scan. Changed. Line 220

You sometimes refer to glutamine side chains instead of glutaminyl side chains (a major context error) Changed

There is a semi-repeat in line 232-233 Corrected

In 3.3 the pH change effects are not well described. The explanation was suggested by the authors. Added.

In line 250 "On the same line" sounds clumsy and colloquial. Changed

In 3.5 RACE is not explained.  Done  

267 - contains your tautology of "extremely crucial". Changed

 289 - Finally. - at the end of the list seems clumsy. Also most of the list seems poorly written. For example Recognised should be "Recognition of" Changed

There are several examples of terms that would benefit from addition of "s" - like induce (induces) in line 282. Corrected

Round 2

Reviewer 2 Report

I feel that the general organisation has improved since the first submission and I have amended my scores accordingly. The manuscript still requires a significant edit in terms of sentence construction and minor points of grammar. I'd suggest running it through an English language expert to iron out the final glitches, which are not vital for understanding the script - but make it a little uneven to read. Items such as the semi-repeat in line 246 about survival factors and the last sentence in 3.4 that does not scan well slightly distract the reader from the quality of central message. The lack of date in the in text refs and the incorrect use of "to" (delete this) in line 52. Also, inappropriate use of apostrophy in "cell's" (line 30).

More importantly (and I think I pointed this out last time) - TG2 acts on peptidyl bound glutaminyl resides. It does not modify glutamine residues (glutamine has a specific meaning relating to a single amino acid not in a peptide bond). This is so central to TG2's activity that though probably a typo - represents a major error that might mislead a reader who is unfamiliar with the enzyme. 

Line 290 needs a comma after "bread". Then "resulting in differential electrophoretic pattern" is guessable in terms of meaning, but is an example of something that could be better described.

In line 305 there is a line "as mentioned in the introduction". If the narrative flowed better there would be no need to say this. The tagged on "extensively" at the end of line 308 does not read well. 

The use of "this functions" in line 250 is grammatically incorrect.

There are quite a few other examples of this kind of thing throughout.

However, I do think that this is an important subject that requires more study - especially amongst the increasing vegan community who are likely to be consuming ever greater amounts of bacterial TG2. I'll suggest acceptance to the editors on the basis that the paper is properly tidied up.

Author Response

Hello reviewer 2

Thanks for the valuable comments and corrections

All corrections are marked in blue.

line 246 was corrected

last sentence in 3.4- was changed .lines 249-250.

date was added. lines 48-49.

To was deleted in line 52

Apostrophy  in line 30 was corrected.

Peptidyle bound glutaminyl resides was corrected in lines: 64, 143-144, 208.

Comma was added in line 289.

The meaning of lines 289-291 was corrected.

Line 306 was corrected.

Line 308 was corrected.

Line 250 was corrected.

English was corrected by native English speaking scientist.

Thanks for upgrading and improving the manuscript